



# The Potsdam Soil Moisture Observatory: High-coverage reference observations at kilometer scale

Elodie Marret[1], Peter M. Grosse[1,2], Lena Scheiffele[1], Katya Dimitrova Petrova[1], Till Francke[1],
Daniel Altdorff[1,4], Maik Heistermann[1], Merlin Schiel[1], Carsten Neumann[5], Daniel Scheffler[5],
Mehdi Saberioon[5], Matthias Kunz[5], Miroslav Zboril[6], Jonas Marach[6], Marcel Reginatto[6],
Anna Balenzano[7], Daniel Rasche[8], Christine Stumpp[9], Benjamin Trost[3], and Sascha E. Oswald[1]

[1]Institute of Environmental Science and Geography, University of Potsdam, Karl-Liebknecht-Straße 24–25, 14476 Potsdam, Germany
[2]UFZ - Helmholtz Centre for Environmental Research GmbH, Dep. Monitoring and Exploration Technologies, Permoserstr. 15, 04318 Leipzig, Germany
[3]Leibniz-Institut für Agrartechnik und Bioökonomie (ATB), Max-Eyth-Allee 100, 14469 Potsdam, Germany
[4]UFZ - Helmholtz Centre for Environmental Research GmbH, Dep. Computational Hydrosystems, Permoserstr. 15, 04318 Leipzig, Germany
[5]GFZ Helmholtz Centre for Geosciences, Section Remote Sensing and Geoinformatics, Telegrafenberg, 14473 Potsdam, Germany
[6]Physikalisch-Technische Bundesanstalt (PTB), Bundesallee 100, 38116 Braunschweig, Germany
[7]Consiglio Nazionale delle Ricerche (CNR), Area della Ricerca di Bari, Via Amendola 122/D-O, 70126 Bari, Italy
[8]GFZ Helmholtz Centre for Geosciences, Section Hydrology, Telegrafenberg, 14473 Potsdam, Germany
[9]BOKU University, Soil Physics and Rural Water Management, Muthgasse 18, 1190 Vienna, Austria

**Correspondence:** Peter M. Grosse (peter.martin.grosse@uni-potsdam.de)
* The first two authors contributed equally to the publication.

**Abstract.**

Cosmic-ray neutron sensing (CRNS) has gained popularity for estimating soil water content (SWC) due to its innovative capability to measure at an intermediate scale—a notable advantage over point-scale sensors, which are often sparsely installed and due to small-scale heterogeneity result in uncertain absolute values. CRNS serves as a crucial link between small and large scales and has been emerging as a reference measurement for remote sensing algorithm validation for its ability to link the small and large scales. Yet, the sparse availability of long-term datasets limits use of this possibility. Within the framework of project SoMMet *21GRD08*, multi-scale soil moisture monitoring was implemented to integrate CRNS with complementary in-situ observations. In this paper, we present harmonized soil moisture data from different sensor types including a CRNS cluster, shallow soil moisture measurements and soil moisture profile data, creating a ready-to-use dataset as reference observation for remote sensing products, covering a highly-instrumented agricultural site in the northeast of Germany. The data include 16 stationary CRNS sensors, with co-located point-scale SWC sensors (mostly permanent), two groundwater observation wells, meteorological records, and data from intensive manual sampling campaigns (covering SWC, bulk density, organic matter, etc.). This dataset distinguishes itself from prior studies by the increased area of approx. 1 km$^2$ while still having a high sensor density and overlapping footprints of CRNS. This allows a reasonable degree of geostatistical interpolation to obtain complete coverage. The data are available under the doi.org/10.23728/b2share.db88e149f7924919be376909856739f1 (Grosse



et al., 2025), providing a new reference data set for remote sensing products, hydrological or land-surface models and for other products linked to soil water balance.

## 1 Introduction

The dynamic storage of water in the soil is an important, but spatially and temporally strongly varying part of the water cycle
and critical for the exchange fluxes between land surface and atmosphere, ecohydrology, agriculture, forestry and groundwater recharge. Consequently, soil moisture has been identified as one of the Essential Climate Variables (Bojinski et al., 2014). Its monitoring is not only related to research from climate science to ecology, but also to applied science on water resources management, flood forecasting, drought monitoring, irrigation management in cropped fields and weather forecasting (Vereecken et al., 2008; Brocca et al., 2017; Moragoda et al., 2022; Levi and Bestelmeyer, 2018; Molenaar et al., 2024; Srivastava, 2017;
Hövel et al., 2025; Pendergrass et al., 2020; Abioye et al., 2020; Lachenmeier et al., 2024; Szilagyi and Franz, 2020). Main challenges are the observation at different scales, limited direct coverage of root-zone soil moisture and deep storage in soils, and uncertainties and discrepancies between satellite remote sensing measurements (Oswald et al., 2024). Accurate measurements for spatial scales beyond the point scale continue to be challenging as small-scale variability caused by heterogeneity in precipitation, soil properties, vegetation, and other influences question the applicability of point-scale sensors to achieve
good spatial and temporal coverage beyond the spot measurement (Peng et al., 2021). Point-scale observations range from campaign-based manual soil sampling to long-term observations using dielectric permittivity-based methods (time-domain reflectometry, frequency-domain reflectometry, capacitance-based sensors) or suction-based methods. These techniques provide observations at high temporal resolution but due to their small support volume (typically just a few cubic centimeters) at very specific locations (Robinson et al., 2008)) and the above-mentioned heterogeneity of soil water content (SWC), their
representativeness remains limited. Combined in networks they offer the possibility to observe representative field-scale soil moisture data (Bogena et al., 2022; Rosenbaum et al., 2012), however, to upscale the point-scale data to an area representative soil moisture is challenging due to the resource-intensive nature of these networks and the conflict with soil management in agricultural settings, where soil moisture observations are of high interest.

Over the past decades, satellite-based remote sensing has significantly advanced the measurement of SWC from space.
The most notable progress has been achieved using microwave sensors, which offer the distinct advantage of being operable under all weather conditions and during both day and night. However, remote sensing estimates of SWC refer to the upper few centimeters of the soil and are acquired at temporal intervals determined by the satellite's observation schedule. Global-scale SWC measurements with spatial resolutions of approximately $25 \times 25$ km$^2$ or coarser, and high temporal frequency (every 2–3 days), are currently available e.g. through missions such as the European Space Agency's Soil Moisture and Ocean
Salinity (SMOS) mission (Kerr et al., 2001), NASA's Soil Moisture Active Passive (SMAP) mission (Entekhabi et al., 2010). Furthermore, the long-term Copernicus program, via the Sentinel-1 (S-1) radar satellites, is fostering the development of operational SWC monitoring at higher spatial resolutions, down to 1 km (Bauer-Marschallinger et al., 2019). The quality assessment of satellite-derived SWC products heavily depends on ground-based measurements (Dorigo et al., 2021). However,





a critical challenge remains: the scale mismatch between the spatial resolution of ground reference data and that of satellite
observations, which poses a significant barrier to the accurate validation of remote sensing products (Gruber et al., 2020).

Introduced in 2008 (Zreda et al., 2008), Cosmic-Ray Neutron Sensing (CRNS) technology has emerged and meanwhile proven to be a valuable method for intermediate-scale soil moisture measurements, applicable in different land-use and vegetation settings, including forest. The non-invasive detector retrieves soil moisture time series by detecting the intensity of epithermal neutrons present above ground. This epithermal neutron count rate is inversely correlated to soil moisture, or more
precisely, the hydrogen content within its support volume, termed "sensor footprint" in this context, extending vertically about 15 to 60 cm and horizontally about 150 to 200 m radius (Köhli et al., 2015; Schrön et al., 2017). The neutron counts are typically accumulated over a period of hours, corrections for other factors such as air pressure applied, and converted to volumetric water content using a custom calibration function (Desilets et al., 2010; Zreda et al., 2012). Significant advancements have been made in the past decade in areas such as signal correction (McJannet and Desilets, 2023), interpretation (Köhli et al., 2021;
Schrön et al., 2023; Rasche et al., 2021), sensor calibration/uncertainty reduction (Schrön et al., 2017), and data processing (Power et al., 2021b). As a result, Cosmic-Ray Neutron Sensor technology is increasingly utilized for diverse applications, besides soil moisture content measurements also for biomass estimation (Jakobi et al., 2018; Brogi et al., 2022), and observations of snow water equivalent (Schattan et al., 2017). However, its major advantage - providing a large-scale and integrative soil moisture measurement within the critical root zone - remains underutilized so far. This can largely be attributed to the
relative sparseness of sufficiently-long records, and the required expertise in their conversion to soil moisture. Nevertheless, despite all the corrections and advancements, the basic ability of the sensor was there from the start - the very large advantage of measuring non-invasively, averaging out the effects of small-scale heterogeneities in its signal.

The non-invasive and low-maintenance CRNS technology, combined with its ability to record data over long periods, made it a promising tool for remote sensing assessments (Babaeian et al., 2018; Mengen et al., 2023; Beale et al., 2021). While
various CRNS-datasets have been published (e.g. Bogena et al., 2022; Cooper et al., 2021; Dorigo et al., 2021), they are mostly composed of solitary, i.e. spatially detached, stations. Although single-footprint sizes represent a major step toward matching the resolution of remote sensing products, they still fall short of the spatial resolution required by some applications. Combining multiple CRNS-sensors in close vicinity remedies this issue. By establishing dense clusters, recent studies have begun to address the lack of comprehensive datasets: these CRNS sensors are operated with overlapping or adjacent footprints
providing a robust option for capturing the spatial and temporal distribution of soil moisture. To date, three data sets have been published. At the pre-alpine agro-sylvo-pastoral headwater catchment site in Fendt, 24 CRNS sensors were installed over a $1\,\mathrm{km}^2$ area for a short period of two months (Fersch et al., 2020). In 2020, at the Wüstebach catchment in the Eifel Mountains, 15 detectors were operated over $0.4\,\mathrm{km}^2$ of forested land temporary for three months (Heistermann et al., 2022). More recently, Heistermann et al. (2023) reported the data from three years of observations of 15 CRNS sensors over a $0.1\,\mathrm{km}^2$
lowland agricultural site at the Leibniz-Institut für Agrartechnik und Bioökonomie (ATB) Marquardt. The latter cluster was the only one designed for a longer-term operation, i.e. more than a year, and specifically has aimed for being able to account for local heterogeneities by its unusual high-density of CRNS detectors.

The current study combines the scale of coverage of the temporary Fendt site investigation with the continuity of observation of the ATB Marquardt site. It was implemented on remaining parts of the former CRNS cluster at the latter site and extended to an area that is larger by an order of magnitude. By that it does not only cover now the complete ATB site, but goes beyond and even includes areas within two communes being part of the City of Potsdam. That is what we will refer to here as the *Potsdam Soil Moisture Observatory* (PoSMO).

The new cluster, installed on a well-controlled agricultural field with on-site weather records, was designed to be able to provide a reference soil moisture dataset for satellite remote sensing. Compared to Heistermann et al. (2022), it extends instrumentation to 1 km² to match the resolution of highly-resolved satellite remote sensing products, maintains a continuous time series since 2019 (seven sensors from the previous cluster) to better capture hydrological extremes, and adds complementary datasets (hyperspectral and Lidar data, stable isotopes, electrical resisitvity tompgraphy, etc.) to enhance soil moisture reconstruction at the km² scale. Besides the soil moisture from CRNS, point-scale soil moisture profiles from 10 cm down to 1 m were measured in combination with the CRNS sensors and these were supplemented by even shallower point-scale soil moisture measurements, aligning with the penetration depth of remote sensing products and reflecting the standard commonly used for their validation.

## 2 Study Area and Instrumentation

### 2.1 General description

In 2019, the University of Potsdam began operating a dense CRNS cluster at the Potsdam-Marquardt site as part of the Cosmic Sense project, named the "Marquardt cluster". This site, located in northeastern Germany in the northwest of Potsdam, has since then been expanded as a key long-term soil moisture monitoring observatory. The major part of the PoSMO is now distributed over the whole research site operated by the Leibniz Institute for Agricultural Engineering and Bioeconomy (ATB), but some stations cover areas outside the ATB site and a few are located completely outside of it. Recent investigations have been part of DFG research unit *Cosmic Sense* and the interdisciplinary EURAMET (European Association of National Metrology Institutes) project *21GRD08 SoMMet* running until end of 2025 and September 2025, respectively. The main aim has been the observation and harmonization of soil moisture across multiple spatial and temporal scales. The automated observations are scheduled to be continued beyond the projects' time frames, so that the published dataset could be updated in the future.

The PoSMO consists of an inner area of about 0.7 km² area at 40 m a.s.l., including cereals, maize, alfalfa, sunflower, rape seed, meadows, diverse types of orchards (cherry, apple, maroni) and young poplars—under rain-fed management. With exception of a few irrigated orchards and local irrigation measures (on potatoes, maize and spring wheat) the site is under rain-fed management. It features 20 agricultural field plots and is situated on predominantly periglacial sandy soils (68–91%), with silt content ranging from 8–27% and clay from 0.6–4.4%. Organic matter content ranges from 0.4–17.3%. The climate is temperate, with an average annual precipitation of 584 mm (1981-2010) and an average annual temperature of 9.3°C (Cfb by Köppen-Geiger). The groundwater table lies between 1.5 and 10 m beneath the surface, and two observation wells installed

in 2019 continue to provide data. Two weather stations, that is one on-site and one located 12 km away by the German
Meteorological service (DWD), ensure comprehensive meteorological observations.

The site is representative of typical lowland agricultural landscapes in northern Central Europe, situated within a climatic
transition zone between maritime and continental influences. Moderate seasonal contrasts allow the site to capture a broad range
of hydro-meteorological conditions relevant to temperate agro-ecosystems. As such, the site serves as a valuable model envi-
roment for investigating soil moisture dynamics across diverse hydro-climatological conditions, including dry periods, snow
events, and variable precipitation patterns. Its setting enhances the transferability of findings to similar agricultural contexts
across the region.

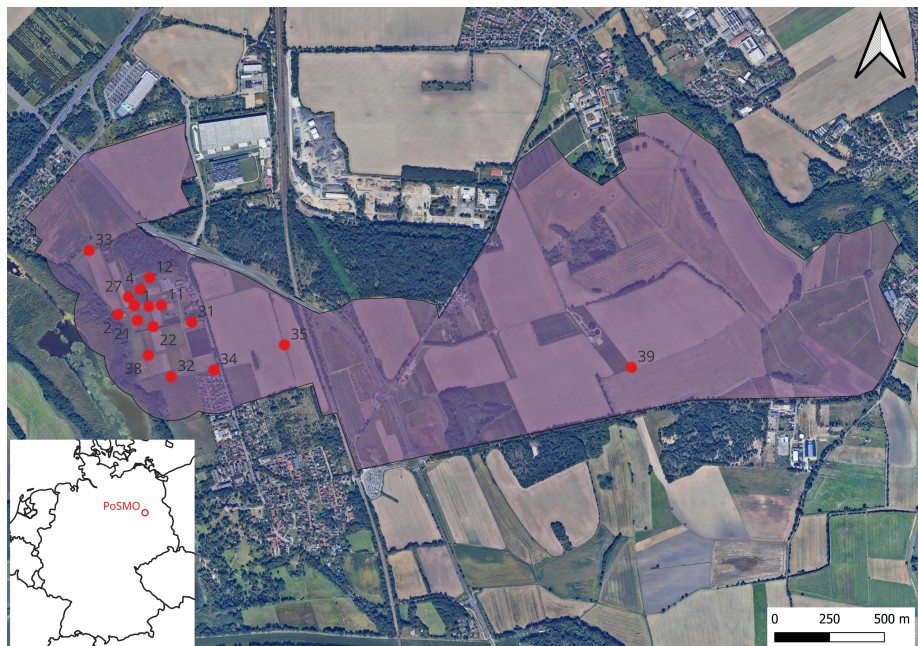

**Figure 1.** Locations of cluster stations (CRNS, shallow SWC, SWC profile) at the Potsdam soil moisture observatory. The total area of mixed
agricultural land use that the observatory represents could be taken as the purple area (size of 3.4 km$^2$, similar land use, soil type and aligning
with the spatial extent of remote sensing products, excluding forest and settlement areas). The background image is taken from Google Earth:
Marquardt, retrieved in September 2023 (Gorelick et al., 2017) , Maxar Technologies (© Google Earth).

## 2.2  Highlights of PoSMO

The Potsdam soil moisture observatory (PoSMO) stands out for several reasons:

– With 15 CRNS stations distributed over an area of approximately 0.7 km$^2$, it covers an area seven times as large as the
previous CRNS cluster. The CRNS locations were placed in order to monitor both vertical and horizontal soil moisture
variability, with one additional CRNS location placed 2 km east to extend the spatial coverage even further. Similar land

use and soil types render the sensor highly representative of the broader surroundings at the resolution of remote sensing products (excluding forested area and settlements). Altogether PoSMO could be taken to represent an area of about 3 km$^2$ of predominantly cropped fields and meadows, with some other agricultural land use and loose urban settlings (Fig. 1).

– PoSMO has been operational in its present form since mid December 2023, allowing observations under varied conditions, including snow events (e.g., snow covered the site for eight days in February 2025), drought, irrigation, and heavy rainfall. Monitoring is intended to be continued, to finally be able to provide a three-years data set (as a future update to the first data set presented in the current paper).

– The test site incorporates diverse land uses and features—including croplands, roads, buildings, though dominated by agricultural area, providing an ideal setup for assessing soil moisture dynamics under contrasting conditions.

– Extensive instrumentation complements the CRNS data. Two SMT100 TDT sensors (Truebner) were installed in December 2023 at each of the 16 cluster locations at 5 and 15 cm depth (totalling to 32 sensors), adding to the 20 soil moisture profile measurements (including PR2/4, PR2/6, ThetaProbes ML2x, TDR) that cover depths from 10 down to 100 or 200 cm. Faulty sensors were regularly replaced to maximize data continuity.

– Soil moisture measurements include a rich dataset of manual and automated readings. In 2023, six calibration campaigns were conducted (in May, July, October and November), covering 145 locations. Manual FDR measurements were collected at six depths, with 33 locations including full soil sampling. In 2024, in four additional campaigns (in April, August and October), 138 new FDR samples and 32 soil bulk density sampling points were taken.

– Soil bulk density is a key parameter for deriving volumetric soil moisture from CRNS observations, just as it is essential for retrieving soil moisture from saturation- or index-based remote sensing products, for hydrological applications, and for simulating CRNS signals in the field. Bulk density values were interpolated from 136 locations at PoSMO including campaigns since 2019 (cf. Fig. 6).

– Larger snow cover events have also been monitored. Thus, a dedicated snow campaign on the 17 February 2025 included 240 snow height measurements, 10 snow bulk density samples, UAV imaging, and wildlife camera deployment to monitor melting.

– To unravel the relationship between soil moisture dynamics and downward soil water fluxes, bulk soil water samples were analysed for stable water isotopes on three occasions and interpretation was aided by dedicated measurements of saturated hydraulic conductivity in the root zone. Measurements were distributed to capture spatial heterogeneity, stemming from the mixed land use covers.

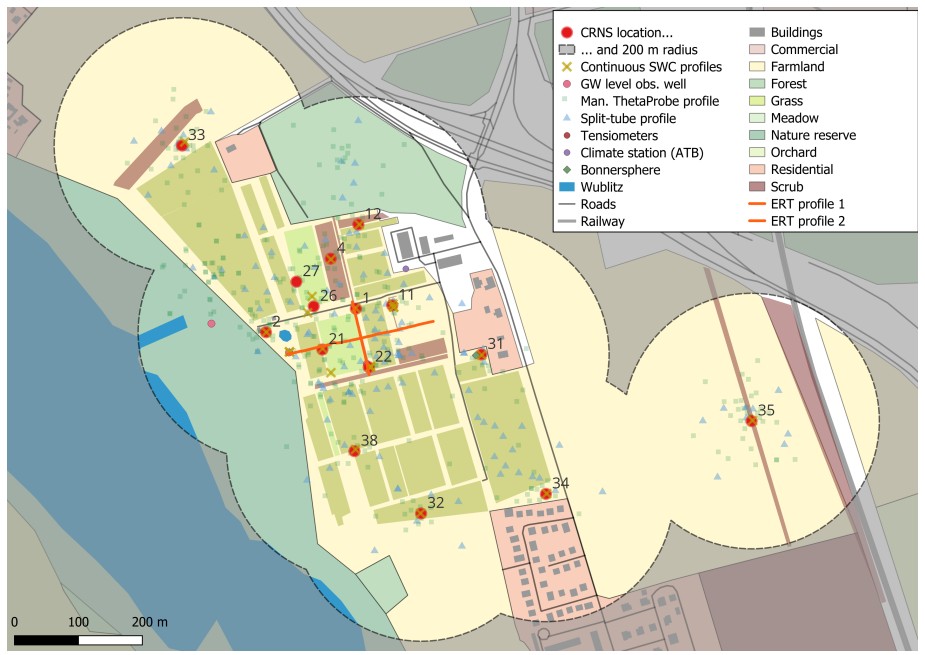

**Figure 2.** Overview of the Potsdam soil moisture observatory central area, showing continuously operating stations (CRNS, SWC profiles, groundwater monitoring, tensiometers, and climate stations), the two ERT transects, hyperspectral camera position, Bonner sphere spectrometer location and calibration sampling points. Background layers illustrate land use such as farmland, buildings, and forested areas etc. are based on OpenStreetMap (OSM) with few amendments (© OpenStreetMap contributors 2025. Distributed under the Open Data Commons Open Database License (ODbL) v1.0. ).

## 3 Observations and data

The following subsections provide an overview of the instruments and observations that comprise the dataset. Table 1 summarizes the various sub-datasets collected from 2023 to 2025, during which period also five drought events (all CRNS sensors providing soil moisture < 15 %) and three extreme precipitation events were observed (daily precipitation over 20 mm). The core components of the dataset include the distributed CRNS stations, vertical soil water content profiles and shallow soil water content time series, remote sensing data, all of which are described in detail in Section 3.1 and listed in Table 1.

### 3.1 Soil moisture from stationary CRNS

From December 2023 to April 2025, we collected epithermal neutron count data from 16 stationary Cosmic-Ray Neutron Sensors (CRNS) with overlapping footprints at the soil moisture observatory established in Potsdam. The locations of the CRNS sensors are depicted in Fig. 1 and 2. The placement of the sensors was informed by several key considerations: (i) to ensure a substantial overlap of the footprints, (ii) to capture the site's variability along the hillslope gradient, (iii) to position some



**Table 1.** Overview of section 3: brief summary of each data subset, main observed variables and units, temporal coverage. Specific details can be found in the subsections and the json files which documents each data subset in the repository. The text in bold indicates the name of the folder with these data sets in the repository, which will be preceded by the number on the left.

| Sect. | Data subset | Main observation variables (units) | Temporal coverage |
|---|---|---|---|
| 3.1 | 16 permanent **stationary CRNS** detectors recorded epithermal neutron counts along with meteorological variables | neutron count rate (cph), air pressure (hPa) and temperature (°C), relative humidity (%) | Dec 2023-Jul 2025 |
| 3.2 | One **muon** detector to represent temporal variability of incoming fast neutrons, as potential correction for CRNS data | muon count rate (cph) | Feb 2023-Jul 2025 |
| 3.3 | **Neutron spectrum** recorded as daily values by a group of outdoor Bonner spheres, as potential correction for CRNS data and detector response | histogram (cph) | Jun - Jul 2025 |
| 3.4 | **SWC profiles** time series at 20 locations (dielectric measurements) | Permittivity (-), SWC ($m^3/m^3$) | Dec 2023-Jul 2025 |
| 3.5 | **Shallow SWC** time series at 16 locations (via TDT) | Permittivity (-), SWC ($m^3/m^3$), Soil temperature (°C) | Dec 2023-Jul 2025 |
| 3.6 | Multiple campaigns with **manual soil sampling** of the upper 30 cm of the soil (split-tubes, ThetaProbes, lab analysis) | Permittivity, SWC ($m^3/m^3$), bulk density ($g/cm^3$), SOM (g/g), texture | May 2023-Oct 2024 |
| 3.7 | **Hyperspectral reflectance** time series on-ground | Reflectance (-) | Oct 2022-May 2024 |
| 3.8 | **Hyperspectral airborne imagery** at one date | Reflectance (-) | Sep 2024 |
| 3.9 | **Airborne Lidar imagery** at one date | Digital Surface and Terrain Model (m), | Sep 2024 |
| 3.10 | **Landuse**, especially crop cycles | sowing andharvest dates | Apr 2021-Jul 2025 |
| 3.11 | **Snow** depth monitoring and some snow sampling (various techniques), areal imagery mosaics from an UAS overflight | Snow water equivalent (SWE) (mm), snow depth (cm) and density ($g/cm^3$); RGB mosaic | Feb 2025 |
| 3.12 | **Groundwater level** and lake level time series | GW level (m.a.s.l.) and distance to surface (m) | Dec 2023-Jun 2025 |
| 3.13 | **Stable water isotopes** measured in soils and groundwater along the hillslope | delta values for $^2H$ and $^{18}O$ | May 2023-May 2025 |
| 3.14 | Permeameter measurements to determine **soil saturated hydraulic conductivity** | Ksat (cm/min) | single days in 2023 and 2024 |
| 3.15 | **Electrical Resistivity Tomography** - two ERT transects | apparent electrical resistivity (ohm m) | Nov 2022 |
| 3.16 | **Long-term CRNS time series** of locations in core area operated during 2020-2025 continuously, thus providing also data during 2023 | Neutron counts, air pressure (hPa) and temperature (°C), relative humidity (%) | 2020-2025 |



sensors near the groundwater well, and (iv) to avoid interfering with agricultural management activities, which necessitated not to directly place sensor on cropped.

The detailed specifications of the sensor cluster are provided in Table 2, including the manufacturer, sensor type, and sensitivity (for further details, see Heistermann et al. (2023)). A total of 16 sensors from various manufacturers were included in the study: five devices from Hydroinnova LLC (Albuquerque, USA), four from Quaesta Instruments LLC (Tucson, USA), four from StyX Neutronica GmbH (Mannheim, Germany), one Canberra (now Mirion Technologies), one from Finapp S.r.l. (San Pietro in Cariano, Italy), and one from Lab-C LLC (Sheridan, USA), which is now associated with Quaesta Instruments.

The sensitivity of each device was determined through parallel measurements with a reference CRNS sensor (calibrator) over a test period, allowing for a comparison of the neutron count rates observed by different instruments. As detailed in Table 2, the majority of the detectors employ gases with high neutron cross-sections to detect neutrons, including $^3$He gas (CRS-1000, CRS-2000) and $^{10}$BF$_3$ enriched gas (CRS-1000B, CRS-2000-B, B-E1-4). The 'HydroSense Dual' detector is based on a multiwire proportional chamber with solid $^6$Li (Fersch et al., 2020; Patrignani et al., 2021), while the StX-140-5-15 sensors utilize

$^{10}$B-lined converters. Finally, the FINAPP5 instrumental detection of neutrons relies on a multi-layer zinc sulfide and $^{100}$Ag doped scintillator with $^6$Li fluoride powder (Gianessi et al., 2024). All of the devices described above are epithermal neutron detectors. Most sensors record also temperature, relative humidity, and barometric pressure using an external sensor. These data were used to correct the raw neutron count rates. For sensors without own external sensor, data from the reference station at location ID 11 were used for corrections.

To exemplify the use of the CRNS observations we have processed the CRNS data to obtain a soil moisture product (Figure 3, see also section 5), which is also included in the data set. Soil moisture estimates derived from CRNS data are susceptible to various sources of uncertainty, including the influence of other hydrogen pools, which are not explicitly accounted for in this SWC series presented in this paper, but implicitly addressed by the local calibration procedure.

Figure 3 presents a comprehensive overview of the time series data from December 2023 to 2025. This heat map reveals
five major drought events (occurring in summer 2024 and summer 2025) and two notable snow fall events in February 2025. The temporal dynamics of the 16 locations exhibit consistent patterns over time. To facilitate a straightforward analysis, the sensors have been grouped according to their geographical location, ranging from the west to the east. It should be noted that, if the effect of snow coverage on the CRNS measurements is evident, this was not a focus on this dataset. For the snow episode documented during winter 2024/2025, the snow coverage, the height, and density measurements are independently detailed
in this dataset (section 3.11). The precipitation data from the DWD weather station (see section 4) demonstrate the CRNS response to rainfall events.

Figure 4 shows two examples of the spatial distribution of root-zone soil moisture as inferred from the CRNS data on 8 September 2024 and 24 November 2024. Spatial interpolation on a 2 m resolution grid takes into account CRNS horizontal sensitivity combined with inverse distance weighting. Among the various sensors, a notable consistency in response patterns can be observed for both wet and dry periods.



**Table 2.** Properties of CRNS sensors used in the Potsdam cluster (including manufacturer, model, and detector mechanism). Also provided is the ratio of the sensor's raw counts of epithermal neutrons to the counts of a calibrator sensor (consistent with Heistermann et al. (2023)), referred to as *sensitivity*. This is a device specific property, which is independent of the site. It can be used to infer a local calibration parameter value (cf. Heistermann et al., 2024)). Some CRNS devices have been refurbished and are technically different to the formerly used device, and thus the sensitivity value had to be updated and is different to the former one. CRNS devices that have been continued at their former location with identical set-up are printed with bold ID. Finapp s.r.l. has changed its CRNS model names; Finapp 3 (F3) refers now to Finapp SWC, Finapp 5 (F5) to Finapp SWC plus, and Finapp 6 (F6) to Finapp SWC Premium. Some Lab-C CRNS devices are currently only provided via Quaesta Instruments, and then the manufacturer listed is Quaesta Instruments.

| ID | Manufacturer | CRNS type | Detection mech. | Sensitivity | remark |
|---|---|---|---|---|---|
| **1** | Hydroinnova | CRS 2000-B | $^{10}BF_3$ gas | 1.191 | |
| 2 | Canberra | self-assembled | $^3$He gas | 0.668 | refurbished; set-up with horizontal tube position |
| **4** | Lab-C | HydroSense dual | $^6$Li foil | 4.537 | |
| **11** | Quaesta Instr. | dual BF3-C-4 | $^{10}BF_3$ gas | 4.871 | |
| 12 | StyX Neutronica | StX-140-5-15 | $^{10}$B-lined | 1.127 | refurbished |
| **21** | Hydroinnova | CRS 2000-B | $^{10}BF_3$ gas | 1.149 | |
| **22** | Hydroinnova | CRS 2000-B | $^{10}BF_3$ gas | 1.163 | |
| **26** | Quaesta Instr. | B-E1-4 | $^{10}BF_3$ gas | 2.487 | single tube of a dual system |
| **27** | Quaesta Instr. | B-E1-4 | $^{10}BF_3$ gas | 2.468 | single tube of a dual system |
| 31 | StyX Neutronica | StX-140-5-15 | $^{10}$B-lined | 2.773 | |
| 32 | Finapp s.r.l. | Finapp-SWC Plus | $^6$Li-doted scinti. | 1.400 | muon sensor included |
| 33 | Quaesta Instr. | BF3-A-3, dual | $^{10}BF_3$ gas | 1.490 | |
| 34 | Hydroinnova | CRS 1000 | $^3$He gas | 0.448 | |
| 35 | Hydroinnova | CRS 1000 | $^3$He gas | 0.689 | |
| 38 | StyX Neutronica | StX-140-5-15 | $^{10}$B-lined | 2.425 | |
| 39 | StyX Neutronica | similar to SP2 | $^{10}$B-lined | 2.236 | refurbished |
| 11val | Finapp s.r.l. | Finapp-SWC Premium | $^6$Li-doted scinti. | 0.873 | not part of the cluster, as testing option |

### 3.2 Additional muon observation

Neutron monitors, such as the the one at Jungfraujoch (JUNG) in Switzerland, are conventionally used to correct locally measured epithermal neutron counts for variations in incoming neutrons. Recent studies suggest that local measurements of muon counts could enhance methods for correcting CRNS data to account for incoming neutron variability (Stevanato et al., 2022). To provide this capability and explore this option, one CRNS sensor operated in the Potsdam cluster (#32) included sensors for detecting muons (Gianessi et al., 2024), and these data are available from January 2023 to May 2025; see also Table 2.

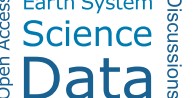

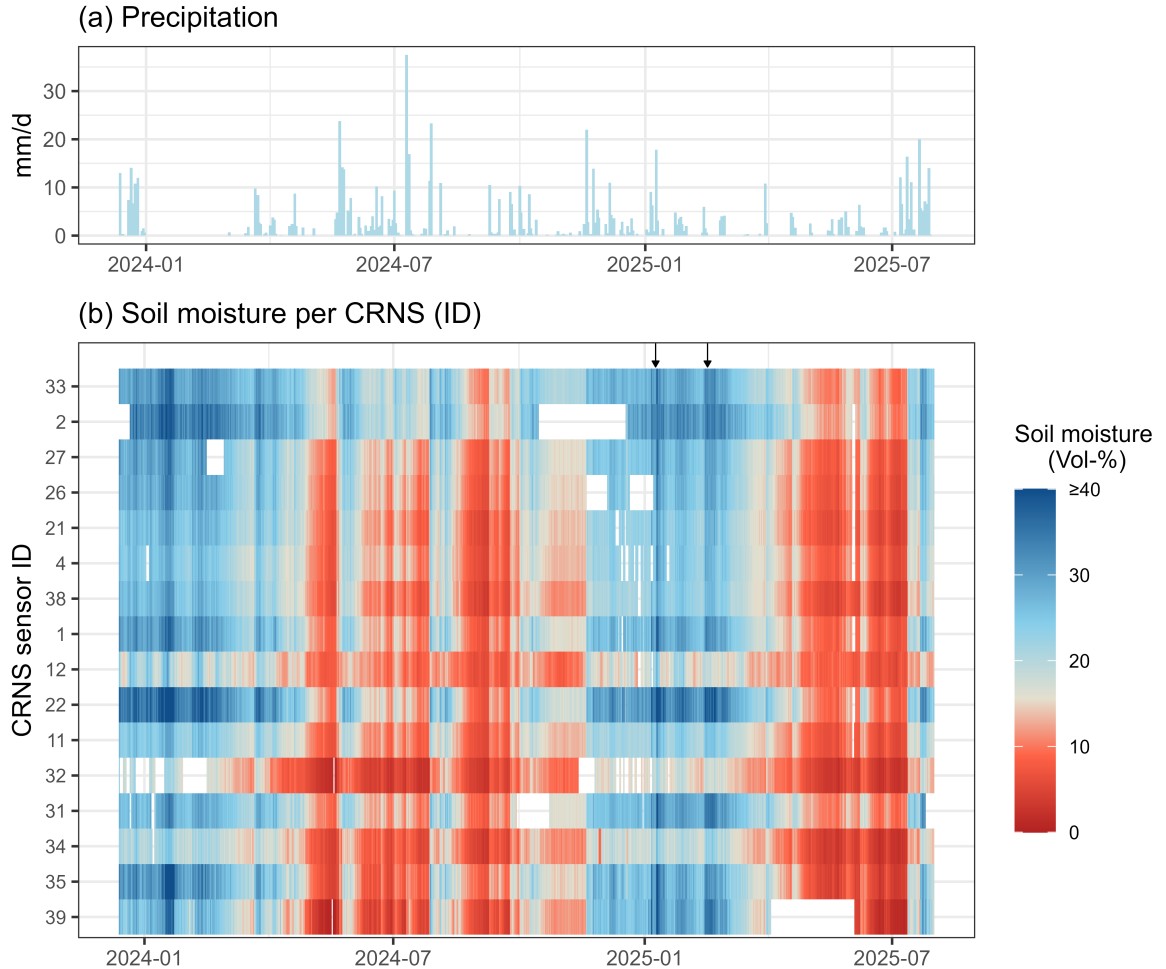

**Figure 3.** (a) Daily precipitation; (b) Root-zone soil moisture time series derived from CRNS measurements, arranged from west to east according to sensor locations. The arrows above the soil moisture time series indicate periods of snow fall that may influence soil moisture dynamics.

### 3.3 Bonner sphere measurements of full neutron energy spectrum

An extended range Bonner sphere spectrometer (ERBSS) consists of a set of moderating spheres of different diameters and a
thermal neutron detector that is placed at the center of each sphere. A typical set has standard spheres made of polyethylene plus a few modified spheres with metal shells embedded in the polyethylene spheres. Each sphere plus thermal neutron detector combination has a different energy response to neutrons. For the standard spheres, the peak of the response function shifts to higher neutron energies as the size of the moderator is increased. For the modified spheres, the response increases dramatically for neutron energies above ∼50 MeV. In addition, it is usual practice to measure also with the thermal neutron detector without





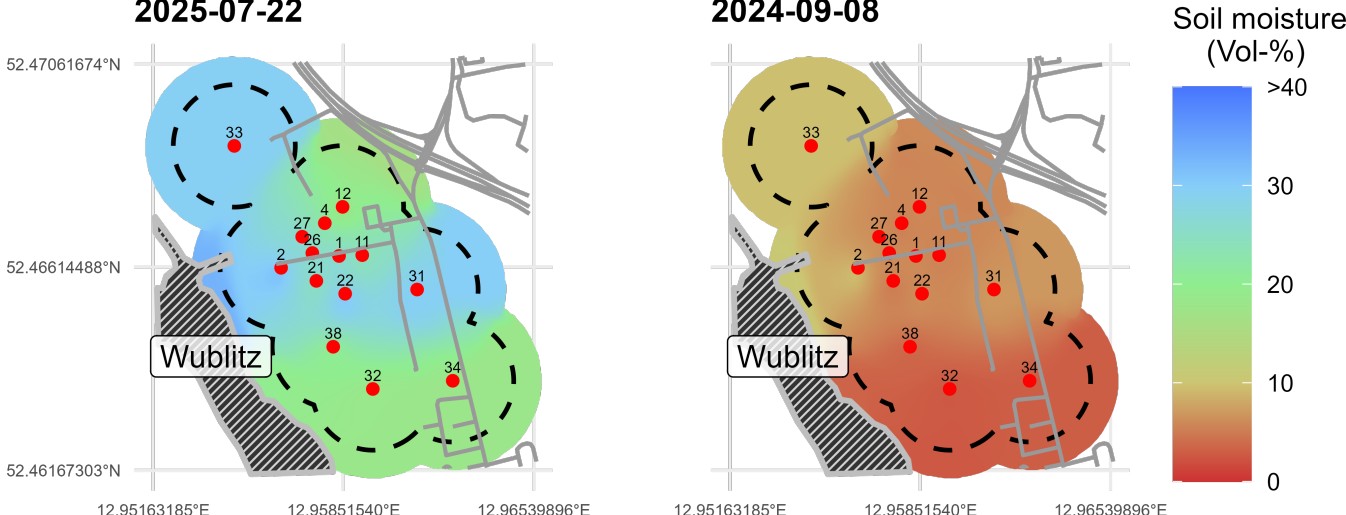

**Figure 4.** Root-zone soil moisture distribution for a relatively moist period of the time series in July 2025(left) and for a late summer day of a drier period in September 2024 (right). Root-zone soil moisture distribution based on the daily soil moisture product derived from CRNS stations within the central area of the Observatory (section 3.1). Dashed circles outline the 150 m footprint of the sensors. Map background information see Figure 2 (© OpenStreetMap contributors 2025. Distributed under the Open Data Commons Open Database License (ODbL) v1.0. )

a moderating sphere (i.e., the bare detector). Any combination of sphere and thermal neutron detector is usually called a "sphere" and this terminology has also been extended to the bare detector.

The Physikalisch-Technische Bundesanstalt (PTB) has been operating its ERBBS system, known as NEMUS Wiegel and Alevra (2002), for about two decades. Recently, the PTB developed an SI-traceable copy of the NEMUS system specifically for automated neutron spectrometry measurements under outdoor conditions. The system NEMUS-UMW (from the German 220 word *Umwelt* - environment) consists of a new subset of NEMUS spheres: 6 polyethylene ones with diameters of 3, 4, 5, 6, 8 and 10 inches, a bare thermal neutron detector and four modified spheres with lead and copper shells. The central thermal neutron detectors are spherical $^3$He-filled proportional counters of the type SP9, manufactured by Centronic Ltd. The specific SP9 counters used in the NEMUS-UMW system were selected for low-level neutron measurements due to their low intrinsic background. The intrinsic (instrumental) background of each SP9 counter was systematically tested and quantified in the 225 "neutron-free" environment of the PTB underground laboratory UDO II. The electronics for neutron signal processing and telemetry of the NEMUS-UMW system are comprised of components that correspond to the state of the art in the CRNS community. Using unfolding procedures Reginatto (2010), one can determine the neutron spectrum from the measured data and the spectrometer response functions. The NEMUS response functions were characterized and validated in the PTB's neutron reference fields, and the NEMUS system serves as a secondary metrological transfer standard for neutron fluence rate 230 measurements. This means that the resulting unfolded neutron spectrum is determined in absolute terms, in units of neutron fluence $\Phi(E)$ in neutrons $\mathrm{cm}^{-2}$.

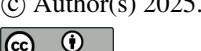



Since 18 June 2025, the NEMUS-UMW system has been operated at the Potsdam site, at a location with permanent power supply as well as close to one CRNS station (#31), shown in Figure 5 (left). The objective of this study is twofold: firstly, to obtain time series of neutron count rate for each spectrometer sphere, and secondly, to provide unfolded neutron spectra that

have been averaged over periods of several hours on different dates and times. Figure 5 (right) shows a time series for the neutron count rates of the spheres, indicating also the precipitation for the days of the measurements as well as four unfolded neutron spectra. The first pair of unfolded spectra was obtained from measurements taken prior to a precipitation event that occurred on 12 July 2025. The second pair of spectra was obtained two days after this event. The unfolded neutron spectra provide detailed information about the energy distributions of neutrons detected on site, as well as potential changes in the

individual energy domains of the distributions. The unfolding solutions for both pairs of unfolded neutron spectra demonstrate stability both before and after the precipitation event. Furthermore, a change in the neutron distribution is evident before and after the precipitation event which can be attributed to the change of soil moisture in the area. The higher moderation of neutrons is attributable to the higher abundance of water (and its hydrogen atoms) in the soil following precipitation, resulting in a shift of neutron fluence from epithermal to thermal neutrons, while the high-energy part of the neutron distribution

remain unchanged. This demonstrates the mechanism underlying the CRNS observation of water content via the detection of epithermal neutron intensities. Furthermore, it shows the potential of a BSS system to validate the neutron data time series measured with neighbouring CRNS detectors.

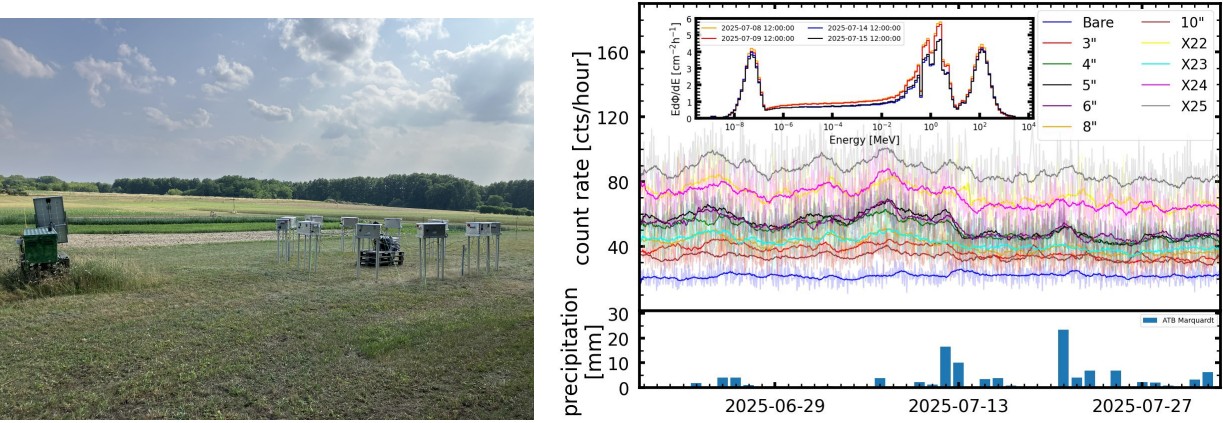

**Figure 5.** Left: Setup for the neutron measurements using the PTB extended range Bonner sphere spectrometer NEMUS-UMW. The individual Bonner spheres are encased in metal boxes to protect them from adverse weather conditions. Each sphere is suspended from springs to minimize noise due to vibrations. The boxes are distributed on a circle of radius of 3 meters, with the data logger and telemetry modules stored in a central box. The spheres are located 1.20 m above ground. Right: Neutron count rates of NEMUS-UMW system for different sphere sizes and thus neutron energy ranges (top figure), along with the precipitation records for the days of the measurements (bottom figure). Inlet graph in the top figure shows four unfolded neutron spectra corresponding to different days denoted in the inlet figure, two before and two after the precipitation event on 12 July 2025.





### 3.4 Soil moisture profiles at point scale

We employed four different methods for measuring soil moisture profiles, each involving different depths and sensors (resulting
in a total of 25 individual profiles, covering about two years of data, Fig. 1). This approach enabled us to capture detailed data on
the vertical soil moisture dynamics relevant to infiltration and drying processes, which are vital for extracting and understanding
CRNS-derived soil moisture measurements (c.f. Scheiffele et al., 2020). The impedance-based profile probes (2 PR2/4 and 13
PR2/6, Delta-T Devices LLC, Cambridge, UK) collected data at 10, 20, 30, and 40 cm depths (with the PR2/6 extending to
60 and 100 cm). Five additional profiles each comprising 4–5 individual impedance-based probes (ThetaProbe ML2x, Delta-T
Devices LLC, Cambridge, UK) extended the network by measuring at depths of up to 200 cm in areas where deeper monitoring
was desirable or where profile probes were impractical to install. Furthermore, five soil moisture profiles were equipped with
five TDR probes each (TDR100, Campbell Scientific Ltd., UK) installed at depths of 9, 11, 25, 45, and 75 cm. Measurements
collected at 15-minute intervals were aggregated to hourly resolution. A two-point calibration (air, water) was applied for the
impedance probes to adjust the raw sensor data. For converting permittivity values into volumetric soil moisture we utilized
the equation according to Zhao et al. (2016) with customized coefficients, as described in Heistermann et al. (2023).

### 3.5 Shallow soil moisture from point-scale sensors

To enhance the monitoring of the upper soil layers at all CRNS stations, additional SMT100 sensors (Truebner, Germany) were
installed at depths of 5 and 15 cm; via determining the oscillation frequency of a time-domain transmission system (Jackisch
et al., 2020) they observe dielectric permittivity and also measure soil temperature. The 5 cm measurement is closest to the
surface soil moisture as observed by satellite remote sensing. Shallower depths than that are prone to larger errors in respect to
a realistic estimate of depth below a usually rough soil surface and the vertical averaging of soil moisture point-scale sensors
such as the SMT100.

Thus, these observations constitute a cluster of soil moisture observation time series that provide a value close to the soil
surface, give a difference (and gradient) in soil moisture for each time and furthermore could be linked to the soil moisture
profiles recorded at the same locations but at different depths (section 3.4).

### 3.6 Campaign based soil moisture for CRNS calibration

To facilitate the calibration of the CRNS stations, a series of extensive field campaigns were conducted at 283 new profile
locations, complemented by the data of two previous campaigns in 2019 and 2022 (Heistermann et al., 2023). This dataset en-
compasses a total of 38 campaigns within the CRNS cluster footprint, with both ThetaProbe and soil core measurements taken
down to a depth of 35 cm. In total, the ThetaProbe profiles were conducted at 447 locations, while the soil core measurements
were carried out at 152 locations, thereby increasing the spatial density of soil moisture observations. All sampling positions
were accurately recorded using differential GPS (dGPS), and the bulk density distribution is depicted in Figure 6, revealing a
relatively homogeneous site. To investigate the potential impact of agricultural practices, such as labor, on soil bulk density, a
test in a plot with maize was done in 2024 to assess potential temporal changes by harvest and tillage (by disc). Bulk density



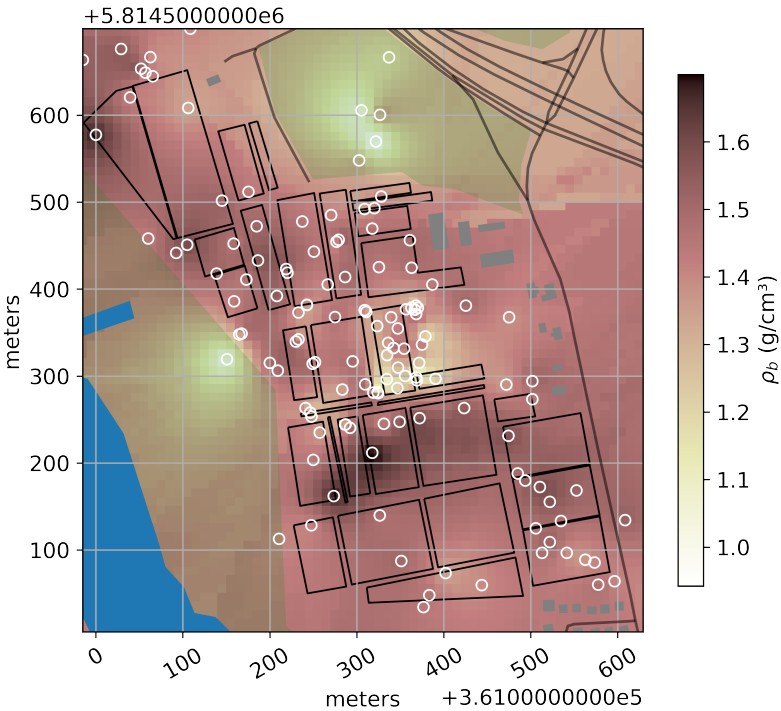

**Figure 6.** Average soil dry bulk density (g cm$^{-3}$) in the top 30 cm of the soil, across the central area of the Potsdam Soil Moisture Observatory; the spatial distribution was obtained by Ordinary Kriging (exponential variogram with a range of 50 m).

was sampled at 0, 5, 10 and 15 cm before and after harvest and tillage events, with a time difference of about 2 months. The results gave no evidence that soil bulk density was different after harvest and tillage events than before, as a Tukeys-HSD test gave a p-value not being significant for rejecting the null hypothesis of being bulk densities being equal before and after. Soil bulk density was also determined specifically at 5 cm depth for ten of the cluster locations, including the more distant one to the East (#39), by taking horizontally cylinder rings of 100 cm$^3$ volume and measuring dry weight after drying in the oven

at 105°C for 24 hours. Values ranged between 1.10 and 1.47 g cm$^{-3}$ with a mean of 1.3 g cm$^{-3}$ and standard deviation of 0.12 g cm$^{-3}$. In accordance with the procedures outlined by Heistermann et al. (2023), key soil properties were derived from oven-dried samples, including dry soil bulk density and water content, as well as from loss-on-ignition analysis for organic matter content and lattice water. Additionally, manual soil moisture measurements were conducted using portable ThetaProbes, employing sensor-specific calibrations and a site-specific conversion of permittivity to soil moisture, resulting in a root mean

square error (RMSE) of ± 0.05 m$^3$ m$^{-3}$ for the soil moisture estimates.





### 3.7 Ground-based reference measurements of Hyperspectral Surface Reflectance

The Hyperspectral Pointable System for Terrestrial and Aquatic Radiometry (HYPSTAR®) is a two-axis, autonomous hyper-spectral radiometer designed to deliver multi-angular measurements of land and water surface reflectance. On 11 October 2022, a HYPSTAR®-XR unit was installed at the ATB HYPERNETS site at (52°27′59.40" N, 12°57′35.16" E), mounted on a 5-meter mast with a 5-meter horizontal boom. The boom was oriented southward toward a bare soil surface to avoid obstructions in the sensor's field of view. The system acquires measurements every 30 minutes between 09:00 and 17:00 UTC across a range of zenith and azimuth angles. It incorporates two spectrometers: a VNIR unit covering 380–1000 nm with 1,330 channels at 3 nm full width at half maximum (FWHM), and a SWIR unit covering 1000–1700 nm with 220 channels at 10 nm FWHM. An onboard camera provides contextual imagery, while an internal LED source tracks calibration stability (De Vis et al., 2024; Goyens et al., 2021). Data are transmitted to a central server and processed using the `hypernets_processor`, a Python-based pipeline for radiometric calibration, quality control, and computation of radiance, irradiance, and surface reflectance. The system rigorously propagates both random and systematic uncertainties, along with spectro-temporal covariance, ensuring that the data meet Fiducial Reference Measurement (FRM) standards. The dataset collected at the ATB Marquardt site includes raw and processed surface reflectance data. The data include a complete time series of angular hyperspectral surface reflectance for a bare soil target under varying illumination and atmospheric conditions. The dataset supports applications such as validation of radiative transfer models and soil spectral analysis. Future work will explore the use of this dataset for estimating surface soil moisture, using both empirical spectral indices and physically based retrieval approaches.

### 3.8 Airborne hyperspectral data

Hyperspectral data were not only observed as time series for a reference area at the ATB Marquardt, within the Potsdam Soil Moisture Observatory, but accompanied by one time airborne campaign on 5 September 2024 providing hyperspectral and a Lidar imagery. The former will be presented in this section, and the latter in the following section 3.9.

**Airborne Hyperspectral Image Cube**

Airborne hyperspectral data were acquired using two pushbroom imaging spectrometers: the CASI-1500 (covering the visible to near-infrared range, VNIR: 380–1050 nm) and the SASI-600 (covering the shortwave infrared range, SWIR: 950–2450 nm), both manufactured by Itres. The sensors were mounted aboard a Cessna 208B aircraft as part of an aerial survey campaign. The native spatial resolution achieved was 25 cm for CASI-1500 and 63 cm for SASI-600. The CASI sensor provided 15 spectral bands at an average spectral resolution of 38 nm, whereas SASI delivered 100 bands with a spectral resolution of 15 nm. The survey consisted of two parallel flight lines, each 375 m swath width, to ensure complete coverage of the designated test site.

**Pre-Processing Workflow for airborne campaign**

Initial data pre-processing was conducted by CZglobe. Radiometric calibration was performed using the RadCorr software package, following the procedures outlined in Hanuš et al. (2023), and employed laboratory-derived sensor calibration parameters. Additional corrections included scattered light correction, frame shift smear correction, second-order light correction, and bad pixel interpolation to ensure optimal signal integrity. Georeferencing was accomplished using a GNSS/IMU navigation

system in combination with a high-resolution digital terrain model (DTM), implemented within the GeoCor software suite. The

325 hyperspectral imagery was projected into the UTM coordinate system (Zone 33N, ETRS-89 datum). Subsequent atmospheric correction and reflectance retrieval were carried out using ATCOR-4 (version 7.1) (Richter and Schläpfer, 2011), developed by ReSe Applications in collaboration with DLR. The algorithm employs the MODTRAN radiative transfer model to correct for atmospheric effects and retrieve surface-level reflectance. The resulting reflectance values are stored as integer-scaled values, where a reflectance of 10.00 % is represented as 1000 (i.e., scaled by a factor of 100).

**Coregistration and final data product for airborne hyperspectral imagery**

Following atmospheric correction, the VNIR and SWIR datasets were spatially co-registered using the AROSICS software (Scheffler et al., 2017). The registration process utilized spectrally overlapping bands at 1004.8 nm (VNIR) and 1002.5 nm (SWIR) to ensure high alignment accuracy. The overall registration error could be reduced from an RMSE of 39 cm to 16 cm. Redundant overlapping bands were removed, and the SWIR data were resampled to match the spatial resolution of the CASI

sensor (25 cm). The final dataset was generated by mosaicking the two flight lines and saved as Cloud Optimized GeoTIFF (COG), resulting in a geometrically and radiometrically consistent hyperspectral cube for subsequent analysis.

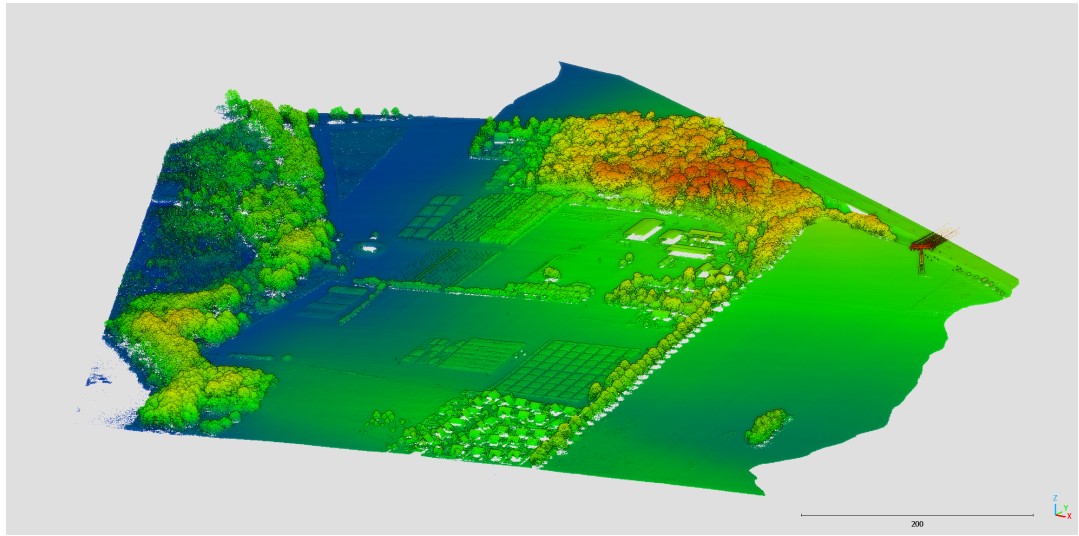

**Figure 7.** Illustration of the lidar data as a height colorized point cloud processed by open source tool CloudCompare, with focus on the central area of the CRNS cluster (14 of 16 stations included); view from South.

## 3.9 Airborne Lidar data

Airborne Lidar (ALS) data was captured using a Riegl LMS Q-780 airborne full-waveform laser scanner mounted onboard a CESSNA 208B Grand Caravan equipped with GNSS/IMU POS AV inertial navigation unit. The site was captured in three

strips, north-south orientation, flown in the morning of 05/09/2024. The Riegl LMS Q-780 operates at a wavelength of 1064 nm with a laser pulse repetition rate of up to 400 kHz and a beam divergence of 0.25 mrad. FOV was 60°. Flight height was





620 m and scan line width was 710 m, 1190 m and 1660 m. The captured strips have an average point density of 7.3 points per m$^2$.

Strips were aligned using the RiPROCESS 1.9.2, RiUNITE, and GeoSysManager 2.2.4 software from RIEGL Laser Mea-
surement System GmbH. For the flight trajectory calculations, the POSPac 8.7 software was used, followed by trajectory conversion in the Riegl-POFImport 1.8 software. For processing in the RiPROCESS software it was necessary to convert the trajectory in POFImport to *.pofx format. In RiProcess, the reference surfaces were first searched automatically. This was followed by the calculation of deviations between the same reference surfaces on different lines. The least squares method was used to minimize the deviations while adjusting the trajectory parameters for each individual flight line. The laser data was then
re-georeferenced using the adjusted trajectories. The error (standard deviation) of the scan data adjustment was 0.007 m (based on 120.000 tie points). Data was georeferenced to the UTM33N (Universal Transverse Mercator - zone 33N) coordinate sys-tem on ETRS-89 (European Terrestrial Reference System 1989). The resulting laser data was exported as point clouds in LAZ format (v1.4), including the so-called Riegl extra bytes, which associate to each point the information from the full-waveform analysis (amplitude and pulse width). The resulting point density is 15.2 points per m$^2$ in the co-registered point cloud.

On the processed data noise was removed using LAStools software and a classification (surface and terrain) of the point cloud was performed. A DTM (digital terrain model), DSM (digital surface model) and nDSM (normalized digital surface model – elevation map) are provided as individual Geotiff rasters with a spatial resolution of 0.25 m. The provided lidar dataset includes flight strip data (including amplitude, reflectance, pulse width) from the Riegl LMS Q-780 as *.las as well as a mosaic of all co-registered *.laz lines and 1000 m x 1000 m tiles.

**3.10 Land use, cropping and irrigation**

The thermal and epithermal neutron counts recorded by the CRNS detectors are highly sensitive to the vegetation. Remote sens-ing algorithms as well, also require information on biomass coverage for calibration. To facilitate a comprehensive comparison between remote sensing data and CRNS measurements, or to enable in-depth analysis, we provide additional information on crop distributions, types, and harvest values. This complementary data set, provided by the ATB (Leibniz-Institut für Agrartech-
nik und Bioökonomie), complete the airborne lidar data, and enables a more accurate assessment of the hydrogen pools. While no irrigation experiments were conducted during the study period, some crops, including potatoes, maize, and spring wheat, re-ceived irrigation during the summers of 2023 and 2024. Additionally, the orchards (cherry and apple/strawberries/wines grape) and blueberry plantations were regularly irrigated during the spring and summer months. All irrigation events were recorded in the dataset (plots, date, amount)

**3.11 Snow cover campaign winter 2025**

Snow cover at the research site is infrequent, typically occurring every 2-3 years. However, its occasional presence significantly influences the vertical soil water balance, and affects the interpretation of CRNS signals. Additionally, the surface's spectral characteristics are either completely masked or modified, making comprehensive documentation of snow cover periods also essential for remote sensing applications. In mid-February 2025, few days of persistent snow cover with depths reaching a





maximum of 6 cm occurred. We recorded its temporal progression using a stationary wildlife camera (SECACAM HomeVista, VenTrade GmbH, Köln, Germany) capturing hourly images. The spatial distribution and properties of the snow cover were mapped using UAV-based RGB imagery with a Mavic Pro (Da-Jiang Innovations Science and Technology Co., Ltd, China) from survey flights at 100 m flight altitude. The imagery was stitched using Photoscan software (Agisoft LLC, St. Petersburg, Russia) and geo-referenced to 20 cm orthophotos provided by the Federal State of Brandenburg, producing an RGB image with
a 3 cm ground resolution.

Additional manual snow sampling involved 240 measurements of snow depth with a ruler and 10 points for density measurements through cylinder cores or by collecting all snow from a specific area for subsequent weighing in the field.

### 3.12  Groundwater and lake levels

Groundwater levels data complement soil moisture observations as they capture the subsurface response to recharge and storage
changes, linking surface and vadose zone dynamics with aquifer behavior. The depth to the groundwater table at the PoSMO increases from very shallow near the lake in the west towards approximately 10 m below ground in the east of the field site. Groundwater during the operation of the PoSMO has been monitored in two wells along the slope. Additionally between September 2022 and January 2025 the lake level (open water body of the Wublitz) was monitored and is provided here. The upslope well is located in the vicinity of CRNS sensor 22 approximately at the middle of the hillslope (ground elevation
36.17 m a.s.l.), the downslope well close to CRNS sensor 2 (31.14 m a.s.l.) and the lake level is observed in a peripheral ditch in the vicinity of the same CRNS sensor (Fig. 2). The well pipes of 63 mm diameter have a length of 2 m, 4 m and 6 m for the lake, well 02 and well 22, respectively. They are filtered over a length of one meter at the lower end and pressure sensors (Hobo U20L, Onset) were installed to record pressure and temperature at a 30 min interval. In well 22, an additional sensor recorded air pressure and temperature but had to be removed because of sensor failure in May 2024. To correct for air
pressure variations after this time, pressure recorded at the CRNS sensors were utilized. Regular manual measurements of the groundwater heads were used to validate the continuous measurements and exclude a drift in the pressure sensor measurements. Starting in September 2023, groundwater was pumped regularly for sampling stable water isotopes (see section 3.13). The resulting periods with artificial drop in water table were excluded from the data set.

### 3.13  Stable water isotopes in soil and groundwater

Stable water isotopes ($\delta^2$H and $\delta^{18}$) are widely used as natural tracers of water transport processes in the critical zone and across scales (Sprenger et al., 2019; Scandellari et al., 2024). Tracking the isotopic signature of precipitation through the soil profile enables qualitative and quantitative assessment of the timing and spatial variability of infiltration and percolation, and, together with soil moisture data, provides estimates of groundwater recharge (Canet-Martí et al., 2023; Wang et al., 2023; Koeniger et al., 2016). To qualitatively assess vertical percolation rates and their spatial heterogeneity within the PoSMO, we
conducted three field campaigns under contrasting hydrological conditions. In May 2023, during dry soil conditions, profiles at locations 2 and 22 were sampled on 24 May, and at locations 21 and 11 on 26 May; groundwater was sampled subsequently on 31 May. No rainfall occurred between these dates, ensuring consistent conditions across sites. In January 2024, following





several days of snowfall and subsequent snowmelt, soil profiles at locations 2 and 22 were sampled on 25 January, and at 21 and 11 on 26 January, with groundwater sampled on the same day. In each campaign, bulk soil samples were collected along the hillslope at four positions (2, 11, 21, and 22, see Fig. 2), covering the full soil depth from 0 to 150 cm at 10 cm intervals. To capture deeper infiltration of the January snowmelt, soil and groundwater samples were collected at all locations on 2 May 2024, and profiles at positions 21, 22, and 11 were extended to 200 cm for additional sampling.

Groundwater isotopes were analysed from piezometers at positions 2 and 22, with additional samples collected opportunistically between May 2024 and May 2025, generally about once per month.

During sampling, soil samples were sealed in Ziploc® bags (S. C. Johnson & Sons Inc., USA) to minimize evaporative losses and groundwater samples were stored in 10 ml vials. All samples were refrigerated until transport to the isotope laboratory of BOKU University in Vienna (Austria) for analysis of $\delta^2$H and $\delta^{18}$O.

In the laboratory, soil water isotopes were determined using the direct-equilibration method (Wassenaar et al., 2008). The isotopic ratios of the vapour of the soil samples as well as the isotopic ratios of the groundwater samples were determined using a Picarro L2130-i laser isotope analyser (Picarro Inc., Sunnyvale, CA, USA). The isotopic ratios of all samples are expressed in $\delta$‰ units, which describe the relative difference in the ratio of heavy to light isotopes in a water sample with respect to VSMOW (Vienna Standard Mean Ocean Water). To better interpret the isotopic signature profiles of the water, the volumetric moisture content of the soil samples was also calculated. To do this, the gravimetric moisture of the samples was multiplied by the bulk density of the soil ($\rho_b$). For $\rho_b$ we took average values by depth as measured earlier for these locations (Heistermann et al., 2023).

### 3.14 Soil saturated hydraulic conductivity in the root zone

Saturated hydraulic conductivity ($K_{sat}$) governs water flow through soil and controls key processes like infiltration, percolation and groundwater recharge. The spatial variability of soil properties, driven by heterogeneity in structure, texture, and land use, makes $K_{sat}$ difficult to measure directly in the field, so it is often derived in the laboratory or modelled (Durner, 1994; Hohenbrink et al., 2023). The PoSMO dataset overcomes this limitation by providing direct $K_{sat}$ measurements at multiple depths and land uses, offering a more reliable basis for hydrological analyses. $K_{sat}$ was determined using a constant head permeameter (a so called amoozemeter), which measures infiltration rates within an auger hole. During measurement, a constant hydraulic head was maintained in the hole while monitoring the infiltration rate until steady-state flow was achieved. $K_{sat}$ was then calculated using the Glover equation (Amoozegar, 1989). A total of 28 measurements were conducted across different land uses, including grassland, orchards (cherry, berry, and apple trees), arable crops, and hazelnut hedges. Measurements were performed at depths of 12.5, 20 and 35 cm in eight soil profiles, with four of these profiles additionally sampled at 50 cm depth, following the methodology described by Elrick and Reynolds (1992). Data were collected between 6 January 2023 and 7 June 2024 under varying wetness conditions. All $K_{sat}$ estimates derived by this method represent the uppermost 50 cm of the soil column.



## 3.15 Electrical Resistivity Tomography (ERT) Surveys

Two Electrical Resistivity Tomography (ERT) surveys were conducted to provide a first overview of the deeper subsurface conditions at the core area of the Potsdam site. Results can provide information about stratigraphic layering and inhomogeneities as well as the approximate groundwater table depth, helping to understand variations in surface soil moisture patterns. We used a Syscal ISIS system with 48 electrodes (IRIS Instruments, Orléans, France) along two perpendicular profiles: Profile 1 (A – A′) with 5 m electrode spacing (235 m total length) and Profile 2 (B – B') with 2.5 m spacing (117.5 m total length; see Figure 2) using a Wenner array. Electrode positions — including absolute elevations — were recorded with a Differential Global Position System (DGPS) antenna (Leica Zeno GG04, Leica Geosystems AG, Heerbrugg, Switzerland) for topographic correction of the resistivity data. The raw data sets from 16 November 2022 were imported and filtered in ProSys II (IRIS Instruments, Orléans, France). The apparent resistivity pseudo section was inverted in two dimensions using the open source GUI ResIPy (Blanchy et al., 2020) on a fine triangular mesh with convergence reached within two iterations..

## 3.16 CRNS stations with long-term operation

Following the operation of the former, much smaller CRNS cluster at the ATB site Marquardt (Heistermann et al., 2023), few CRNS stations were not deinstalled and instead were integrated in the new cluster. These are CRNS with bold ID in Table 2, and the same holds for the accompanying SWC profile measurements. Thus, they provide a continuous time series from 2019 to 2025, spanning the time frame of the old cluster (1 September 2019 to 30 November 2022), being part of the PoSMO (14 December 2023 to 31 July 2025) as well as the year between clusters. For convenience of data users interested only in this longer term data set, they are provided here as separate data files. The three years of these time series are based on neutron counts and soil moisture profiles as provided Heistermann et al. (2023). The complete neutron count time series is consistently processed, corrected and calibrated as the current data set described in 5 and for the soil moisture profile as described in section 3.4.

## 4 External data

The subsections refers to datasets provided by third parties, either previously published or available through other channels. These datasets are relevant to interpret our data sets presented and thus are considered useful to readers.

## 4.1 DWD/ATB weather station

A climate station equipped with a heated tipping-bucket rain gauge is situated in the north-eastern part of the study area, as depicted in Fig. 2. The station recorded standard climate variables at an hourly resolution, including air temperature, relative humidity, precipitation, soil temperature at multiple depths (5, 10, and 30 cm), and solar irradiation, as well as wind speed and direction. The original data are publicly accessible via the ATB website (ATB Technology Garden, 2025, last accessed on 01 July 2025). This dataset provides valuable information for understanding the climate conditions in the study area, and

is an essential component of the overall dataset presented in this paper. The nearest climate station operated by the German
Weather Service (DWD) is located at Telegrafenberg in Potsdam, approximately 12 km south-east (station ID 03987). The
corresponding data are publicly available through DWD's open data repository (DWD, 2025) (). The nearest climate station
operated by the German Weather Service (DWD) is located at Telegrafenberg in Potsdam, approximately 12 km south-east
(station ID 03987). The corresponding data are publicly available through DWD's open data repository (DWD, 2025).

## 4.2   Incoming Neutron Flux

The incoming cosmic-ray neutron flux is provided by the Neutron Monitor Database at http://www.nmdb.eu (last accessed
on 01 July 2025). In accordance with previous studies by Hawdon et al. (2014); Baatz et al. (2015); Jakobi et al. (2018);
Baroni et al. (2018), the Jungfraujoch (JUNG) neutron monitor is recommended for correcting the incoming neutron flux at
the Potsdam site. Recent approaches include interpolation between different global neutron monitors and promise to be more
accurate at locations in-between neutron monitors (McJannet and Desilets, 2023).

## 4.3   Other spatial data and maps

A digital elevation model (DEM) with a resolution of 1 m × 1 m is available for download from the Landesvermessung und
Geobasisinformation Brandenburg (LGB) website at https://geobroker.geobasis-bb.de (last accessed on 01 July 2025), with an
accuracy of 30 cm. Additionally, a soil map from the state of Brandenburg, BUEK300 (LBGR, 2025), provides soil types and
texture data beyond the PoSMO area at a scale of 1:300 000 and is publicly accessible through the Brandenburg Geoportal at
https://geoportal.brandenburg.de (last accessed on 01 July 2025). For fieldwork and visualization purposes (Fig. 1), we utilized
OpenStreetMap data layers, which are available for download from the Geofabrik website at (last accessed on 01 July 2025).
Specifically, we employed the land use, waterways, and traffic ways data layers, which provide valuable information for spatial
analysis and visualization. Data are available under ODbL license ()

## 5   CRNS-based soil moisture estimation

The period presented in the dataset spans from 14 December 2023 to 31 July 2025. For the data processing of CRNS neutron
count time series the PYthon tool (Power et al., 2021a) was used. The raw neutron counts were filtered for any data points that
deviated by more than 3 standard deviations ($3\sigma$) from the moving average. The neutron counts are influenced by atmospheric
factors; therefore, we applied the following corrections: the effects of barometric pressure were corrected as well as the effects
of incoming cosmic-ray neutron flux, using the station Jungfraujoch (Switzerland), following Zreda et al. (2012). Effects of
atmospheric vapor content were corrected using the method described in Köhli et al. (2021). CRNS time series were calibrated
using the transfer function from Desilets et al. (2010) with parameters $a_0$, $a_1$, and $a_2$, equal to 0.0808, 0.372, and 0.115,
respectively. The estimation of soil moisture from neutron intensity was calibrated by using local soil moisture measurements
in the sensor footprint, weighted following Schrön et al. (2017) 3.6. Resulting soil moisture time series are provided as hourly
intervals and to reduce the statistical noise in CRNS data a 12-hour moving average (center) was applied.
To enhance sensor to sensor comparison, for example to process the neutron counts with a universal $N_0$ (Heistermann et al., 2024), detector sensitivities are provided (Table 2). The sensitivity of each detector was measured using a collocated calibrator probe or another CRNS sensor with known sensitivity, and the ratio of calibrator to stationary neutron counts was used to determine the sensitivity factor (Schrön et al., 2018).

## 6    Summary use cases for the data

The data set presented and described here can contribute to understand soil moisture spatial patterns at landscape scale and related soil moisture dynamics, especially in the first decimeters of soil. But the design of PoSMO goes beyond that, as it closes the spatial gap between point observations, or even soil moisture in-situ networks, and larger scale observations and modelling. In particular satellite remote sensing can use the data as reference for soil moisture related products, while being different to the usual sites for calibration and validation (cal-val). For this purpose, PoSMO was established at the km scale (which is also a typical resolution of current hydrological or also land-surface models). Thus, several grid cells of remote sensing products or distributed land surface models could be directly compared, assimilated or used for validation, as has been done with stand-alone CRNS already (e.g., Baatz et al., 2017; Fatima et al., 2024).

PoSMO provides ready-to-use, high-resolution SWC time series which will facilitate the cross-validation and uncertainty quantification of the increasing number of high resolution (e.g. 1 km) SWC products coming from downscaling coarse-resolution RS microwave sources and reanalysis data or RS active microwave sources (Fan et al., 2025). As an example, Figure 8, on the left, shows the Copernicus SWI product (soil water index) derived from S-1 constellation covering the Potsdam Observatory. On the right, the figure zooms in on the cluster locations with respect to the Copernicus SWI 1 km pixels. The higher support of each CRNS and the dense number of CRNS distributed in the area of 3.4 km$^2$ will enable an extensive comparison between RS and reference SWC measurements, minimizing the spatial representativeness uncertainty of the ground data.

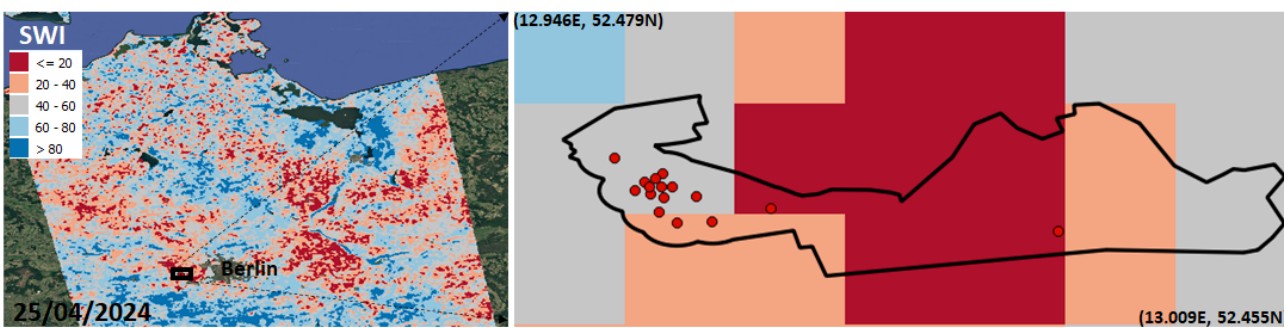

**Figure 8.** Left: Example of Copernicus SWI on 25 April 2024. Right: Zoom into the PoSMO area, with dots indicating an individual cluster location with CRNS, SMT100s and soil moisture profile each. The stations data can be flexibly aggregated and directly assigned to a particular remote sensing pixel, if desired as reference data.

The extensive supplementary data accompanying the soil moisture observations provide the context needed to interpret their spatial and temporal dynamics within a broader hydrological framework, linking them to controlling variables such as soil properties (e.g. saturated hydraulic conductivity or bulk density), as well as supporting the interpretation of water fluxes from
525 stable water isotopes and groundwater data, or exploiting hyperspectral imagery to assess vegetation traits, soil characteristics, and surface conditions.

## 7   Data availability

For data archiving, we utilized EUDAT (https://eudat.eu). The data resides on the B2share server (doi.org/10.23728/b2share. db88e149f7924919be376909856739f1) (Grosse et al., 2025). Its structure corresponds to the subsections of this paper. Each
530 data subset is accompanied by a metadata text file (JSON format), offering additional details on the data format.

*Author contributions.*  SO, PG, TF, LS, and KDP designed the study and coordinated the instrumentation; PG, EM coordinated the data management; EM and PG ed the writing of the manuscript; PG was responsible for setting up and maintaining the instrumentation; MS assisted with data analysis; BT co-designed the instrument network and provided data on land use, yields and irrigation water use; CN and DS contributed the airborne hyperspectral imagery, MS the hyperspectral ground-based time series and MK the airborne Lidar imagery; KDP
did experimental work in the field in respect to stable water isotope data and together with CS the laboratory isotope analyses; DR provided soil moisture profiles from TDRs; MZ, JM, SR conceived the Bonner sphere set-up and performed the measurements with help of PG; SO headed the PoSMO effort and is the principal investigator of this study; all authors contributed to writing particular parts of the manuscript.

*Competing interests.*  No conflict of interest to declare.

*Acknowledgements.*  This research was funded by the Deutsche Forschungsgemeinschaft (DFG, German Research Foundation) – research
unit FOR 2694 "Cosmic Sense", project number 357874777, and project SoMMet that has received funding from the European Partnership on Metrology (Funder name: European Partnership on Metrology, Funder ID: 10.13039/100019599, Grant number: 21GRD08 SoMMet), co-financed from the European Union's Horizon Europe Research and Innovation Programme and by the Participating States.

We thank the Helmholtz-Centre for Environmental Research - UFZ, Leipzig, for providing the majority of CRNS detectors deployed in this cluster. Furthermore, we acknowledge the help of our technician Peter Bíró and student workers for assistance in field work and data
preparation: Isabel von Boetticher, Paola Aleman Serrano, Jakob Terschlüßen, Milan Flügel; and project work by Sebastian Rothermel. The measurements with a hyperspectral airborne sensor was partly based on use of Large Research Infrastructure CzeCOS supported by the Ministry of Education, Youth and Sports of CR within the CzeCOS program, grant number LM2023048



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
