# Peer review of "The Potsdam Soil Moisture Observatory: High-coverage reference observations at kilometer scale"

_Earth System Science Data, 2025_

## Author Comment (AC1)

**Interactive Discussion: Author Response to Referee #1**

**The Potsdam Soil Moisture Observatory: High-coverage reference observations at kilometer scale**

Elodie Marret, Peter M. Grosse et al. Earth Syst. Sci. Data Discuss., doi:10.5194/essd-2025-546

**RC:** Reviewer Comment, AR: Author Response, ☐ Manuscript text

Dear Madam or Sir,

thank you very much for your referee report and for the time and effort you invested in reviewing our manuscript, as well as for the helpful suggestions to improve its clarity and grammar. We sincerely apologize for the grammatical issues in the previous version and will carefully revise the entire manuscript for language and readability.

Please find below our point-by-point responses. We plan to address all your comments in the revised version of the manuscript.

Thank you again for your valuable feedback and your support of this process.

Kind regards,

Peter Martin Grosse (on behalf of the author team)

RC: L5 - suggest "... for remote sensing algorithm validation due to its ability..."

AR: Will be implemented.

RC: L15 - suggest "The data are available from: https://doi.org/"

AR: Will be implemented.

RC: L19 – rewrite the first sentence it is very hard to get past without rereading many times. Do you mean "Soil water storage varies spatially and temporally and is critical for understanding the water cycle, fluxes between the land surface and atmosphere..."?

AR: We gratefully accept the suggestion for clarifying the sentence. .

RC: L21 - what is an essential climate variable as defined by Bojinski? Sentence needs explanation.

AR: We will add the sentence "An Essential Climate Variable (ECV) refers to a physical, chemical, or biological variable, or a set of interconnected variables, that is crucial for defining Earth's climate."

RC: L25 - Fix grammar "The main challenges in soil moisture observation are...???"

AR: Will be corrected.

RC: L34 – suggest deleting "at very specific locations"

AR: We will change the sentence to "These techniques provide observations at high temporal resolution but due to their small support volume (typically just a few cubic centimeters) at selected locations (Robinson et al., 2008)) only. Thus, due to the aforementioned heterogeneity of soil water content (SWC), their representativeness remains limited.

RC: L52 - suggest changing to "...technology has proven to be a valuable method for intermediate..."

AR: Will be implemented.

RC: L58 – suggest "Neutron counts are typically accumulated over several hours, corrected for factors such as air pressure, and then converted to volumetric water content using a custom calibration function."

AR: Suggestion accepted.

RC: Fig 1 - can you make the site numbers a different colour like white so they can be read

AR: Will be implemented.

RC: L138 – are the TDT/FDR measurements field calibrated or are factory default calibration used? This is important as factory default values can be very poor

AR: The TDT/FDR sensors were field calibrated, as stated in L258–260. The respective information will be added to L138.

RC: L173 – the instruments sold by Quaesta are made under licence from Hydroinnova – i.e. they are the same thing. It might be a different model but it's the same technology

AR: To our knowledge, Hydroinnova supplies its systems with a Quaesta Logger as part of the complete setup. We operate different systems utilizing 3He, Li, and BF3. Following its merger with Lab-C, Quaesta now distributes systems under its own name. As far as we are aware, all three brands — Hydroinnova, Quaesta/Lab-C, and Lab-C — use Quaesta Loggers.

RC: L185 – what correction approaches (pressure, vapour, intensity) and calibration equation have been applied to get soil moisture? NOTE – I now see this section 4 (maybe add a note that it is coming later)

AR: We will add a forward reference indicating that the correction and calibration procedures are described in detail in Section 5.

RC: L200 – A figure comparing the relative intensity of a couple of adjacent neutron sensors would be nice to see if they respond similarly

AR: Our focus here is on the derived soil moisture, not on the raw count intensity, as the (dis)similarity between stations can be better interpreted on the scale of soil moisture units. Comparative neutron count data are already presented in Heistermann et al. (2023), Fig. 6 showing the similar response of sensors at the site. In the current paper, we provide raw neutron counts in the data files as well as sensor sensitivities in table 2 for interested readers.

RC: L227 – fix reference

AR: Will be fixed.

RC: L453 – were not removed?

AR: Suggestion accepted.

**RC:** L499 – what does 3.6 mean?

AR: The value 3.6 was a typo and will be removed.

**References**

Heistermann, M., Francke, T., Scheiffele, L., Dimitrova Petrova, K., Budach, C., Schrön, M., Trost, B., Rasche, D., Güntner, A., Döpper, V., Förster, M., Köhli, M., Angermann, L., Antonoglou, N., Zude-Sasse, M., and Oswald, S. E.: Three years of soil moisture observations by a dense cosmic-ray neutron sensing cluster at an agricultural research site in north-east Germany, Earth System Science Data, 15, 3243–3262, 10.5194/essd-2025-54610.5194/essd-15-3243-2023, 2023.

Robinson, D. A., Campbell, C. S., Hopmans, J. W., Hornbuckle, B. K., Jones, S. B., Knight, R., Ogden, F., Selker, J., and Wendroth, O.: Soil Moisture Measurement for Ecological and Hydrological Watershed-Scale Observatories: A Review, Vadose Zone Journal, 7, 358–389, 10.5194/essd-2025-54610.2136/vzj2007.0143, 2008.

---

## Author Comment (AC2)

**Interactive Discussion: Author Response to Referee #4**

**The Potsdam Soil Moisture Observatory: High-coverage reference observations at kilometer scale**

Elodie Marret, Peter M. Grosse et al. Earth Syst. Sci. Data Discuss., doi:10.5194/essd-2025-546

**RC:** Reviewer Comment, AR: Author Response, ☐ Manuscript text

Dear Wolfgang Wagner,

thank you very much for your referee report and for the time and effort you invested in reviewing our manuscript, as well as for the helpful suggestions to improve its clarity and grammar.

Please find below our point-by-point responses. We plan to address all your comments in the revised version of the manuscript.

Thank you again for your valuable feedback and your support of this process.

Kind regards,

Peter Martin Grosse (on behalf of the author team)

- RC: The paper primarily is a comprehensive description of all data collected at the Potsdam Soil Moisture Observatory, and while reading it I felt a certain fatigue towards the end of the paper. To improve engagement of the readers, I recommend incorporating more visualizations to highlight the properties and quality of the different datasets. Additionally, including some exploratory results could make the paper more compelling.
- AR: In response to the reviewers' feedback, the manuscript will be restructured and we will incorporate your feedback as well as some additional figures.
- RC: Section 6 discusses the validation of the Copernicus SWI product as a use case. However, while the fused ASCAT-Sentinel-1 1 km SWI product is sampled at 1 km (hence its acronym 1km SWI), its effective spatial resolution is much coarser (5–15 km; see, e.g., [DOI: 10.1016/j.jhydrol.2022.128462]). Instead, it would be more appropriate to reference e.g. the 1 km Copernicus Sentinel-1 surface soil moisture (1km SSM) product, which not only has fine sampling but also a high resolution of 1km.
- AR: We agree with the reviewer. In Figure 8, the 1 km Copernicus Sentinel-1 surface soil moisture (1km SSM) product was already shown. We correctly addressed the Copernicus SSM product in the text and in the caption.
- RC: As noted by other reviewers, the English language needs improvement, particularly in the introductory sections.
- AR: We sincerely apologize for the linguistic shortcomings. We thank the reviewer for this constructive comment and have thoroughly revised the manuscript to improve clarity, grammar, and readability, particularly in the Introduction.

- RC: In line with RC2's feedback, the section on the Bonner sphere appears out of place and should be rewritten to fit better with the rest of the text.
- AR: We will rewrite and shorten this part.
- RC: Figure 3: Why is the intense precipitation event in summer 2024 (> 30 mm/d) not visible in the CRNS time series?
- AR: The CRNS-derived soil moisture time series, as shown in Figure 1, does reflect the intense precipitation event in summer 2024. The reason this event is not easily visible in Figure 3 is that the soil moisture data there are aggregated at daily resolution.
- RC: Figure 3: It would be nice to see also normalized soil moisture values to better compare the time series.
- AR: The primary aim of this paper is to provide the dataset and enable further analyses. We agree that normalized soil moisture values would facilitate comparison of the time series and acknowledge this as a valuable suggestion for future use of the dataset.
- RC: Line 227: What is an "unfolding procedure"?
- AR: The unfolding procedure involves reconstructing the neutron energy spectrum from the measured neutron count rates of the Bonner spheres through deconvolution of the measurement data. Measurements obtained with a Bonner sphere spectrometer typically provide an indirect rather than direct measurement of the neutron energy spectrum, as the recorded data represent a convolution of the response function with the true spectrum. Deconvolution inverts this process, enabling the determination of the spectrum based on the measurements, the Bonner sphere response functions, and additional relevant experimental information. In this work, we employed the unfolding software MAXED. For a detailed description of the method, readers are referred to Reginatto (2010), as including an extensive explanation here would detract from the focus of the chapter. We term it a "procedure" because we have developed code that automates the unfolding process.
- RC: Line 233: Weird to read here: "The objective of this study is twofold ..."
- AR: Will be rephrased.

**References**

Reginatto, M.: Overview of spectral unfolding techniques and uncertainty estimation, Radiation Measurements, 45, 1323–1329, 10.5194/essd-2025-546https://doi.org/10.1016/j.radmeas.2010.06.016, pROCEED-INGS OF THE 11TH SYMPOSIUM ON NEUTRON AND ION DOSIMETRY, 2010.

Figure 1: Short time series from 2024 showing precipitation (a), CRNS-derived soil moisture (b), and point-scale soil moisture measurements at 5, 10, and 20 cm depth (c, d, e). Grey lines indicate individual measurement locations, and the black line shows the average.

---

## Author Comment (AC3)

**Interactive Discussion: Author Response to Referee #2**

**The Potsdam Soil Moisture Observatory: High-coverage reference observations at kilometer scale**

Elodie Marret, Peter M. Grosse et al. Earth Syst. Sci. Data Discuss., doi:10.5194/essd-2025-546

**RC:** Reviewer Comment, AR: Author Response, ☐ Manuscript text

Dear Madam or Sir,

thank you very much for your referee report and for the time and effort you invested in reviewing our manuscript. We also appreciate your positive assessment of the dataset and your constructive suggestions to improve the structure, clarity, and documentation of the paper. We sincerely apologize for the linguistic and structural shortcomings in the previous version.

Please find below our point-by-point responses. We plan to address all your comments in the revised version of the manuscript.

Thank you again for your valuable feedback and for supporting the improvement of our work.

Kind regards,

Peter Martin Grosse (on behalf of the author team)

RC: The Introduction should conclude with a concise summary of the paper's structure and the data presented in this data paper.

AR: We will add a respective summary to the end of the introduction.

RC: L59: There are more papers on signal correction that could be mentioned here, e.g. Baatz et al. (2015) introduced a biomass correction for CRNS, Davies et al. (2022) tested optimal temporal filtering methods for CRNS.

AR: We will add additional references here.

RC: L62: Brogi et al. (2022) is not about biomass estimation using CRNS. I believe you meant Brogi et al. (2025).

AR: Will be corrected.

RC: L63: Here you could also cite Bogena et al. (2020).

AR: We will add the indicated reference.

RC: L65-67: This sentence is unnecessary and could be removed.

AR: Sentence will be removed.

RC: L75: "three such data sets".

- AR: Will be corrected.
- RC: L123: A separate "Highlights" chapter does not appear necessary. Consider integrating its content into the Introduction as part of the motivation.
- AR: Will be implemented.
- RC: L125: This statement is difficult to understand without referring to Figure 2.
- AR: Will be rephrased.
- RC: L137: Refer to Fig. 2.
- AR: Will be implemented.
- RC: L157: Methods and results are mixed in this chapter, which is inappropriate for a scientific publication. Please restructure the manuscript to clearly separate them. You may refer to Heistermann et al. (2022) as a good example, where the methods are presented first, followed by two separate chapters describing the data provided with the paper and exemplary results from the data analysis.
- AR: We thank the reviewer for this constructive comment on the manuscript structure. In the revised version, we will clearly distinguish between the Methods and Results sections to ensure a more coherent and scientifically appropriate organization. The revised structure will follow the example of Heistermann et al. (2022), with the methods presented first, followed by separate sections describing the datasets accompanying the paper and representative results from the data analysis.
- RC: L182–185: Since all CRNS stations are located in close proximity, it would be more appropriate to use meteorological data from the reference station for corrections of all CRNS stations. This approach ensures that corrections are applied consistently, increasing data consistency. Moreover, reference data are generally more accurate than the lower-quality sensor data used at the CRNS stations.
- AR: Our reference station (ID11) is also a CRNS station equipped with the same type of sensor as the others, not a calibrated meteorological station. Using data from this single station for all corrections would not reliably represent potential spatial variations across the site, especially for humidity. Therefore, each CRNS station was equipped with its own external sensors to account for microclimate conditions (e.g., proximity to the Wublitz water body (ID2) or location within a poplar stand (ID4)). This approach ensures that footprint-specific conditions are properly captured.
- RC: L184–188: In agricultural fields, such as those in this study, biomass changes constantly over the years, which can significantly influence CRNS signals depending on the type of vegetation (e.g., Jakobi et al., 2022). Therefore, the calibration will not implicitly account for this effect. Please discuss this aspect.
- AR: We will add discussion on this aspect.
- RC: L208: The chapter on Bonner sphere measurements feels somewhat isolated. It is also rather long and distracts from the main focus of the paper, i.e., soil moisture data. Therefore, this chapter should be shortened and better integrated into the manuscript.
- AR: Will be shortened and better introduced.
- RC: L399: The section on stable water isotopes in soil and groundwater appears off-topic for a paper focused on soil moisture. Given that only three campaigns may not provide sufficient accuracy to infer groundwater recharge, and the paper already covers a wide range of topics, consider removing this part.

- AR: We agree that the collected isotope data alone may not suffice to infer groundwater recharge. However, we'd like to point out that their conjunctive use with the other data sets may open additional options for seepage water-related analyses. Thus, we would like to retain this section and data despite their limitation.
- RC: L514-521: Unfortunately, this example demonstrates that the dataset's value for remote sensing validation is quite limited, as only a few grids of the RS product are actually covered by the CRNS data, with most sensors clustered within a single grid. Therefore, I suggest removing this part.
- AR: Many thanks for the comment, which helps us to clarify this point.

The Potsdam network differs from other CRNS sites (e.g., Bogena et al. (2022)) due to its uniquely high station density within a single 1-km2 area.

A recent study comparing COSMOS data in Germany with 15 widely used satellite soil moisture products Schmidt et al. (2024) showed that even a 1-km satellite resolution does not resolve the spatial scale mismatch: one single CRNS footprint covers <10% of a 1-km grid cell, and individual pixels often include land-cover types outside the CRNS support area (see Fig. 7 in Schmidt et al. (2024)).

Figure 8 demonstrates that the dense configuration at PoSMO significantly improves the spatial representativeness of CRNS measurements within a 1-km pixel and thus supports more robust CRNS—remote sensing comparisons. Moreover, within this specific site we provide an actual measurement representative of the 1-km pixel, and the locations of CRNS stations in the surrounding pixels could potentially allow extending the spatial coverage to approximately 3.4 km² (see Figures 1 and 8). We therefore consider this site a valuable extension to the existing CRNS reference infrastructure.

- RC: Data archive: I downloaded some of the data (e.g., CRNS, profile, muon) to check whether the files are well documented and complete. The README file provides a good overview of the data and the units of the values. However, there is no description of the file formats. In addition, the CRNS data are split across two separate files, which is confusing. The same issue applies to the SWC profile data.
- AR: We will add the description of the file formats to the README file. We deliberately decided to separate CRNS-counts from CRNS-soil moisture (and profile-based permittivity records from converted soil moisture series) because we are convinced that each of the latter provides the most convenient entry point for users interested in the use of the SM-data, while the former offers more methodological improvements for reprocessing the raw data. We do not expect typical use cases to use both (raw and processed data) thus we preferred to keep the respective file smaller with single variables each. In case you refer to the difference between CRNS data and the long-term observations, we acknowledge, this might be confusing. We will rename the parts of the data set and describe in the manuscript as well as the README more clearly the difference between the data. While the stationary CRNS contains all data from the PoSMO since 2023, a subset of CRNS sensors as well as soil moisture profiles provide continuous data since 2019, which we think is valuable to present as a separate data set.
- RC: Figure 1 only shows locations of CRNS stations (not shallow SWC and SWC profile stations).
- AR: The shallow SWC and profile sensors are all located in close proximity of the stations and would not be discernible at the map. We will add this explanation to the figure caption.
- RC: Figure 5: The graph on the right is not easily readable and should be enlarged. The image on the left does not add much value.
- AR: Image will be removed, graph enlarged.

**References**

Bogena, H. R., Schrön, M., Jakobi, J., Ney, P., Zacharias, S., Andreasen, M., Baatz, R., Boorman, D., Duygu, M. B., Eguibar-Galán, M. A., Fersch, B., Franke, T., Geris, J., González Sanchis, M., Kerr, Y., Korf, T., Mengistu, Z., Mialon, A., Nasta, P., Nitychoruk, J., Pisinaras, V., Rasche, D., Rosolem, R., Said, H., Schattan, P., Zreda, M., Achleitner, S., Albentosa-Hernández, E., Akyürek, Z., Blume, T., del Campo, A., Canone, D., Dimitrova-Petrova, K., Evans, J. G., Ferraris, S., Frances, F., Gisolo, D., Güntner, A., Herrmann, F., Iwema, J., Jensen, K. H., Kunstmann, H., Lidón, A., Looms, M. C., Oswald, S., Panagopoulos, A., Patil, A., Power, D., Rebmann, C., Romano, N., Scheiffele, L., Seneviratne, S., Weltin, G., and Vereecken, H.: COSMOS-Europe: a European network of cosmic-ray neutron soil moisture sensors, Earth System Science Data, 14, 1125–1151, 10.5194/essd-2025-54610.5194/essd-14-1125-2022, 2022.

Schmidt, T., Schrön, M., Li, Z., Francke, T., Zacharias, S., Hildebrandt, A., and Peng, J.: Comprehensive quality assessment of satellite- and model-based soil moisture products against the COSMOS network in Germany, Remote Sens. Environ., 301, 113 930, 10.5194/essd-2025-54610.1016/j.rse.2023.113930, 2024.

---

## Author Comment (AC4)

**Interactive Discussion: Author Response to Referee #3**

**The Potsdam Soil Moisture Observatory: High-coverage reference observations at kilometer scale**

Elodie Marret, Peter M. Grosse et al. Earth Syst. Sci. Data Discuss., doi:10.5194/essd-2025-546

RC: Reviewer Comment, AR: Author Response,

Manuscript text

Dear Madam or Sir,

thank you very much for your detailed and constructive referee report, and for the time and effort you dedicated to reviewing our manuscript. We highly appreciate your comments, which helped us identify aspects that required clearer explanation and methodological refinement.

Please find below our point-by-point responses. We plan to address all your comments in the revised version of the manuscript.

Thank you again for your valuable feedback and your support of this process.

Kind regards,

Peter Martin Grosse (on behalf of the author team)

- RC: The color scheme of Fig. 2 could be updated to improve the contrast among vegetation types. It is very difficult to tell them apart in the map.
- AR: The color scheme of Fig. 2 will be revised. Similar vegetation classes will be merged, and color contrasts will be adjusted to enhance readability and improve visual differentiation among the vegetation types.
- RC: Fig. 4 please note the interpolation method.
- AR: In response to the reviewers' feedback, the manuscript will be restructured. Figure 4 will be moved to the Results section. We will add a short sentence on how the image was created (i.e. spatial smoothing of the data for overlapping or between footprints footprints). However, as the CRNS sensors provide soil moisture for a large horizontal footprint, partially overlapping for the PoSMO, they are an actual measurement and not an interpolated soil moisture from a point measurement. We will make this more clear.
- RC: A key limitation of cosmic ray neutron sensors is that their penetration depth is approximate, but this site has the advantage of multi-depth conventional soil moisture sensors. Could a comparison graph between CRNS and conventional sensor be made perhaps similar to Fig. 3, but showing the difference between the CRNS measurements, and the conventionally measured values interpolated to the same nominal depths? This will be very useful in giving the readers an understanding of the measurement uncertainty.
- AR: We agree, that an estimate of the penetration depth in combination with CRNS soil could be helpful to interpret CRNS soil moisture data, in respect of its depth representativeness. We will provide in Figure 3 the estimated penetration depth calculated using point-scale soil moisture. We will also add some discussion on the uncertainty of estimating the penetration depth on sparse point-scale data.

However, we disagree, that a direct comparison of CRNS and point-scale data will help readers to understand the "uncertainty" of the measurement. Such a comparison might mislead readers to account for the point-scale measurements as the "ground-truth". We do not understand your suggestion for a comparison by using "[...] conventionally measured values interpolated to the same nominal depths". Maybe an approach like the exponential filter "aggregated over the corresponding CRNS depths"?

We would rather include in the manuscript a part where to discuss a appropriate comparison between CRNS and sparse point-scale data could be done. This requires in our opinion a more comprehensive assessment, beyond the scope of a data paper.

Following another reviewers' suggestion, we will provide an additional graph showing a time series of CRNS and point-scale soil moisture side-by-side, however, without the aim to show the uncertainty of CRNS, but rather to give space to discuss the need and approaches how to achieve a meaningful comparison between CRNS and sparse point-scale data. This requires in our opinion a more comprehensive assessment, beyond the scope of a data paper.